# Lessons Learned from the Experiences of Domestic Violence Service Providers in Times of Crisis: Insights from a Central Asian Country

**DOI:** 10.3390/ijerph21101326

**Published:** 2024-10-07

**Authors:** Akmaral Karabay, Saltanat Akhmetova, Naureen Durrani

**Affiliations:** 1Graduate School of Education, Nazarbayev University, Astana 01000, Kazakhstan; 2Sociology and Anthropology Department, School of Sciences and Humanities, Nazarbayev University, Astana 01000, Kazakhstan; saltanat.akhmetova@nu.edu.kz

**Keywords:** crisis, domestic violence, health emergency, violence against women, Central Asia, Kazakhstan, service providers

## Abstract

Domestic violence is a widespread problem in both stable and crisis contexts. During crisis-driven periods, such as environmental, economic, political, and health emergencies, existing gender inequalities are exacerbated, and the risks of violence against women (VAW) are amplified. This qualitative study explores the experiences of professionals working in VAW organisations in a Central Asian country during the COVID-19 pandemic. By interviewing 45 professionals from social care organisations in Kazakhstan, this study aims to understand the impact of COVID-19 on the ability of VAW organisations to assist victims of domestic violence and comprehend the adjustments they made to support victims. The findings shed light on the challenges faced by VAW organisations, including reduced capacity, increased service demand, the shift to remote services, and funding cuts. The study highlights the critical role of these organisations in crises and urges the consideration of lessons learned to prevent VAW in emergency and non-emergency situations. In the Central Asian region, where domestic violence is persistent, this research offers valuable insights for interventions during and after crises. The study offers effective strategies for achieving Sustainable Development Goal 5.2, which aims to eliminate violence against women, and SDG 3.8, ensuring access to healthcare, psychological support, and safe environments.

## 1. Introduction

This study examines the perspectives and experiences of professionals in violence against women (VAW) organisations in Kazakhstan, focusing on the impact of the COVID-19 crisis on VAW in the country and the capacity of organisations to assist survivors of domestic violence. Given the importance of addressing VAW as a critical aspect of achieving gender equality and women’s empowerment, the research aims to shed light on how professionals navigate the challenges posed by crisis situations. This investigation is particularly relevant within the framework of the United Nations’ Sustainable Development Goal (SDG) 5.2, which targets the eradication of VAW.

The recent global COVID-19 pandemic is a reminder that violence against women (VAW) is prevalent in both non-crisis and crisis contexts. Crisis-driven periods characterised by environmental disasters, economic turbulence, political volatility, social instability, military conflicts, and public health emergencies reinforce existing gender inequalities and exacerbate the risks of VAW [1,2]. Natural disasters, such as Hurricanes Hugo and Katrina in the USA [3,4], Hurricane Mitch in Honduras and Nicaragua [5], and the tsunami in Sri Lanka [6], have been found to heighten the risks of physical, psychological, and sexual violence against women and girls [7]. In conflict zones, VAW is employed as a weapon for military and political goals [8], as seen in instances in Colombia [9] and the ongoing conflict in Ukraine [10].

While COVID-19 is no longer an emergency, other emergencies are likely to arise in the future. For instance, the spread of diseases, such as the measles outbreak in 2023 across multiple countries [11,12] and the monkeypox epidemic in 2023–2024 [13], raises concerns about an increase in domestic violence, also called intimate partner violence. Although women are not exclusively victims of domestic violence, they bear the predominant burden of intimate partner violence worldwide [14]. As governments work towards containing and mitigating the impacts of these emergencies, they must remain prepared to respond to escalations in domestic violence. It is essential to prioritise the protection of women vulnerable to domestic violence during crises beyond the COVID-19 pandemic. The post-crisis recovery period, characterised by such factors as homelessness, unemployment, and increased alcohol and drug use, is reported to intensify domestic violence [15]. Studies reported that violence often continued or resumed with greater severity after a disaster [16]. Moreover, the impact of domestic violence experienced during crisis periods can have long-term effects on survivors [3]. Therefore, it is crucial to enhance capacities to support survivors for an effective crisis response [17].

Professionals working in VAW organisations, shelters, and social care organisations play a vital role in protecting women and addressing their needs during stable and crisis times [18]. However, during emergencies, such as the COVID-19 pandemic, victims often lose access to immediate support from family and friends, making the role of these organisations even more critical [19]. Shelters, social care organisations, and emergency hotlines became lifelines for domestic violence victims during the recent quarantine [20,21,22]. Yet, these institutions can also face challenges and become weakened during disasters and complex emergencies [19,23].

Studies on the experiences of VAW organisations during crises show that despite the complexities and challenges, they strive to “do more with less” [2] (p. 109). However, recent studies have highlighted reduced capacity and increased service demand during the COVID-19 pandemic [24,25]. Limited resources made it difficult for organisations to maintain their activities and provide the necessary support to domestic violence victims [26,27,28,29,30]. Lockdown restrictions also posed a significant concern, as reduced shelter capacity increased women’s vulnerability to violence [31,32,33]. Many service providers adapted their services by offering online and telephone support while ensuring confidentiality and safety for survivors living with their abusers [31,32]. Yet, access to such sessions remained challenging for women due to the lack of privacy from their abusive partner [34]. Under such pressure caused by the shift to remote services, the well-being of VAW professionals was also affected, leading to burnout, stress, and exhaustion [25,29,32,33,35].

Despite the significant role played by organisations addressing VAW, and domestic violence in particular, there is a lack of research on the perspectives and experiences of professionals working with survivors during crises [31,36,37,38]. Understanding the experiences and lessons learned by these professionals is crucial for comprehending the dynamics of service provision and the challenges faced during emergencies [23,39]. To better understand the experiences of service providers during emergencies, such as COVID-19, this study delves into the experiences and perspectives of VAW professionals to identify the limitations that impacted the organisational capacity to deliver high-quality care during crisis periods for domestic violence survivors. The study was conducted in Kazakhstan, a Central Asian country where domestic violence is a significant concern, even in non-crisis circumstances [40]. However, research on this topic in Central Asia is limited [41], with the exception of Kyrgyzstan [42,43,44], and particularly scarce in Kazakhstan. 

This study included 45 domestic violence professionals working in organisations at the intersection of public health and social care. The study aimed to answer two research questions: (1) What are staff views about the impact of the pandemic on VAW? (2) How did professionals adapt their practices to support domestic violence victims during the pandemic? This study offers valuable insights that can be extrapolated to both future emergency and non-emergency situations, enabling effective strategies to address domestic violence. This knowledge is crucial for developing more effective strategies aligned with Sustainable Development Goal (SDG) 5.2, which aims to eliminate violence against women, and SDG 3.8, access to healthcare, psychological support, and safe environments for domestic violence survivors during crises and beyond. 

## 2. Conceptual and Terminological Overview

Among the various forms of VAW, domestic violence—often overlapping with the concept of intimate partner violence [45]—emerges as the most prominent form due to its frequent association with women’s fatalities globally [46,47]. Intimate partners are responsible for 38% of women’s killings [48]. Framed as a gendered issue, it is associated with the ideology of masculinity, which encompasses the endorsement of patriarchal attitudes, beliefs, and behaviours assigned to men during socialisation [49,50]. Conceptually, domestic or intimate partner violence refers to acts of violence that occur between people who are in, or have had, an intimate relationship in a domestic setting [51]. Such violence encompasses physical, emotional, economic, and sexual abuse, along with intimidation, isolation, and coercion [52]. Terminologically, in contrast to domestic violence, the concept of intimate partner violence offers a gendered lens to understand VAW, narrowing the issue to two individuals in an intimate relationship. Therefore, this study adopted the term domestic violence due to its contextual relevance, particularly in cases where violence is influenced by the interactions between a woman’s husband and her mother-in-law [53] or other family members of the husband. Moreover, since the prevalence and nature of domestic violence varies significantly across regions and cultures, depending on factors such as cultural norms, gender roles, and socioeconomic conditions [54], understanding these variations is crucial for developing effective prevention and intervention strategies. The section below delves into the sociopolitical context of Kazakhstan to situate the analysis within the specific cultural, legal, and societal frameworks that shape the experiences and responses to domestic violence in the country.

## 3. Research Context

### 3.1. The Cultural Construct of Shame in Central Asia

Central Asia comprises Kazakhstan, Kyrgyzstan, Tajikistan, Turkmenistan, and Uzbekistan, with each country having varying degrees of gender inequality and different legal and institutional systems to address domestic violence [40]. In particular, in this region, domestic violence has traditionally been perceived as a private matter and often accepted and justified due to deeply ingrained cultural beliefs and norms of “shame” [44,52]. This concept of shame, referred to as “uyat” in Kyrgyz, Kazakh, and Uzbek cultures and as “sharm” or “ayb” in Tajik [55], represents a comprehensive value system, embodying a long-standing tradition that dictates moral and ethical behaviour within these societies [56]. Upholding these values is often associated with achieving a positive social reputation and receiving societal approval [56], while non-conformity to these societal norms is perceived as shameful [55]. In many instances, this cultural understanding of shame functions as a mechanism to regulate gender roles and penalise behaviours that deviate from societal expectations [57]. 

Following the independence of Central Asian countries, the emphasis on such moral norms and traditions to shape unique national identities has further contributed to gender conservatism and increased VAW, and domestic violence in particular, in Central Asia [58]. For instance, the prevalence of domestic violence reached a 30% abuse rate among women by intimate partners in Kyrgyzstan, 1 in 5 women experienced domestic violence in Tajikistan [59], and 900 female suicides were attributed to domestic violence in 2020 in Uzbekistan [60]. Specifically, in Kazakhstan, 33% of women have experienced intimate partner violence [61], and approximately 400 women die from domestic violence each year [62]. As scholars explain, discussions around shame are still being used in contemporary Central Asia to control female behaviour, thereby reinforcing male authority over women’s mobility and sexual autonomy [55]. This focus on shame ties a family’s honour closely to the modesty of its female members, whose perceived or rumoured inappropriate behaviour tarnishes a family’s reputation [63]. Therefore, women experiencing domestic violence may demonstrate a reluctance to report abuse due to fears of shame and judgment, while also having a lack of awareness about available support services [64,65].

### 3.2. Development of Domestic Violence Legislation in Kazakhstan

Kazakhstan is a secular country with a population of just over 20 million. It is home to more than 130 ethnic groups, with Kazakhs making up around 70%, and Russians about 20%, as the predominant ethnic groups [66]. Since gaining independence in 1991, Kazakhstan has made a number of efforts to address gender inequality and domestic violence, such as establishing the National Commission for Women and Family Affairs in 1998 and domestic violence units within the Ministry of Internal Affairs [67]. The Laws “On the Prevention of Domestic Violence” and “On State Guarantees of Equal Rights and Equal Opportunities for Men and Women” were adopted in 2009 [68]. The Concept on Family and Gender Policy 2030, adopted in 2016, aimed to reduce violence and close gender gaps in political representation, labour participation, and work–life balance [69] (Decree of the President, 2016). At the international level, Kazakhstan ratified treaties such as CEDAW and the 2030 Agenda for Sustainable Development to demonstrate its commitment to reducing gender inequality and VAW and domestic violence in the country. Amidst these efforts, ironically, in July 2017, the Criminal Code was amended to decriminalise “battery” and “intentional infliction of light bodily harm” [70], effectively eliminating the possibility of criminal prosecution for most domestic violence cases. In 2019, Kazakhstan passed the law “On the Prevention of Domestic Violence” [71], which is seen as a significant step towards preventing or reducing the incidence of domestic violence and supporting the broader goals of SDG 5 on gender equality. Although the law neither criminalised VAW nor sought to prevent it in the long term, it focused on short-term measures, such as providing shelter at governmental crisis centres and other social services to a survivor and banning contact between a survivor and an abuser for a month.

To address domestic violence and support survivors, various VAW organisations in Kazakhstan offer a range of resources and services to assist women in difficult situations, such as domestic violence. These organisations are primarily located in urban areas, which leaves victims in rural regions with limited immediate access to social care [70]. As of 2022, there were approximately 42 crisis centres [72], some of which provide shelter while others offer only legal services. Additionally, 21 Mother’s Houses [55] were initiated by a group of entrepreneurs to deliver specialised support for specific groups, such as mothers with newborn infants, to prevent them from abandoning their children. Thus, while only focusing on a specific group of women, Mother’s Houses also provide services to domestic violence victims.

Despite these efforts, domestic violence remains a critical issue. The VAW statistics provided in Figure 1 demonstrate the changes in the number of reported incidents of VAW from 2007 to 2022, with sporadic significant drops in a few years [73]. However, reports of domestic violence specifically have risen steadily since 2007 [74]. According to the Ministry of Internal Affairs, there were 99,026 reports related to family disputes, but only two-thirds of these cases resulted in court sentences with administrative penalties [75]. Moreover, although the number of recorded criminal offences appears low (Figure 1) [76], it is important to note that domestic violence was decriminalised in 2017, removing the prosecution of domestic violence cases as criminal offences. This might reflect challenges in legal processes, reporting, or the societal handling of domestic violence. This implies that many cases go unreported due to policy limitations or are silenced by domestic violence victims, thus skewing the overall data on VAW. Furthermore, a 2017 survey revealed that around 33% of women had experienced abuse by an intimate partner, including various forms of physical and sexual violence [61]. Social norms remain permissive towards VAW, with 74% of men and 63.8% of women agreeing that a man is justified in beating his partner [77]. Furthermore, findings from the 2015 MICS Kazakhstan survey indicated that 1 in 7 women justified domestic violence, suggesting many incidents may go unreported [78].

Unfortunately, the COVID-19 containment measures only escalated the size and magnitude of domestic violence in Kazakhstan. Specifically, the pandemic led to a 21% increase in reported instances of domestic violence across the country [56,79]. This, in turn, garnered significant public attention towards the existing issues of domestic violence and the legislative mechanism that failed to address them [70]. Due to the increased number of survivors seeking help during the quarantine, there was a lack of available places to accommodate them in shelters [70]. Lockdown measures were detrimental to women experiencing violence, limiting their access to health services and psychological support [79]. Vulnerable populations, including migrants, women, persons with disabilities, and children, faced additional challenges in accessing healthcare during the pandemic [79]. The increase in femicide exposed the problem of domestic violence and VAW in general and raised concerns about the inadequate protection provided by society and the government [56], unveiling challenges faced by the legal system in addressing VAW in the region and the country.

Although the emergency period of the global pandemic has ended both globally and in Kazakhstan, domestic violence continues to be a persistent issue. The recent high-profile case in Kazakhstan involving the murder of a wife by her husband, who was an ex-minister of the country, sparked national outrage. This tragedy exposed the prevalence of domestic violence and led to public demands for stronger legal protections. In response, the government passed “Saltanat’s Law” (named after the ex-minister’s wife) in April 2024, reintroducing criminal charges for battery and intentional harm, which had been downgraded to administrative offences in 2017 [80]. The law is designed to promote women’s rights and improve their safety, while also introducing amendments to Kazakhstan’s Criminal Code, the Law on the Prevention of Domestic Violence, the Law on Marriage and Family, and other laws. Although seen as a positive step, experts remain sceptical of the law for its lack of comprehensive measures to protect women and prevent instances of domestic violence [75], necessitating further exploration of support mechanisms provided by law enforcement agencies and VAW organisations.

## 4. Materials and Methods

This study employed a qualitative research design to explore the lived experiences [52] of professionals working in VAW organisations during the COVID-19 emergency. The study addressed two research questions:What are staff views about the impact of the pandemic on domestic violence?How did professionals adapt their practice to support survivors of domestic violence during COVID-19?

### 4.1. Participants and Organisations

Utilising purposeful sampling [81], we reached out to all 63 VAW organisations to recruit professionals working with domestic violence survivors. As a result, we conducted virtual interviews with 45 professionals who consented to take part in the study. All except three participants were women, with ages ranging from 27 to 60 (see Appendix A). Their experience varied from less than 1 year to 26 years. The participants held various positions within their organisations. Twenty-one of the participants held leading positions, such as directors, heads, and chairpersons, 6 were lawyers, and 18 were frontline workers, such as nurses, social workers, and coordinators.

The participants’ organisations varied in ownership, employee count, funding sources, and services provided. Out of the 45 participants, 6 worked in the state sector, 6 in the private sector, and 33 in NGOs. State-owned organisations generally had more employees, while private organisations and NGOs had smaller staff sizes. NGOs and private organisations often relied on external funding, resulting in more unstable and limited budgets. In contrast, government-funded state centres could afford to have permanent staff. The centres are located in 12 out of the 17 administrative regions in Kazakhstan, representing the 3 biggest cities of the country and other cities in northern, central, western, and eastern parts of Kazakhstan. Two organisations offered only legal aid services, while all other organisations provided a full spectrum of services, including free shelter stays, psychological and medical support, legal aid, job training, job search support, and assistance in finding housing and childcare. VAW professionals were also actively involved in prevention efforts, conducting seminars and training in schools, working with children, and promoting hotlines and crisis centres through media and medical organisations.

### 4.2. Instrument and Data Collection

After obtaining approval from the authors’ Institutional Research Ethics Committee (No. 411/20052021), virtual interviews were conducted with the 45 VAW professionals over 9 months, from October 2021 to June 2022. The virtual mode of interviews allowed us to collect data in a relatively short time from a large number of participants who were geographically dispersed and had busy schedules [82]. 

Conducting semi-structured interviews enabled the examination of professionals’ experiences of working with domestic violence victims under COVID-19 preventative measures, providing rich and nuanced insights into their challenges and adaptation strategies. The following areas were covered during the interviews: (a) demographic characteristics of VAW professionals and their organisations, (b) the effects of anti-pandemic measures and lockdowns on the VAW organisations, and (c) challenges that VAW organisations faced.

Although the majority of interviews were conducted individually by the first author, the first four interviews were conducted in pairs, with one researcher serving as the primary interviewer and the other researcher taking notes on emerging themes. This approach was chosen to facilitate a more comfortable and interactive environment for participants, potentially leading to richer data. The note-taker’s role was to capture detailed observations and emerging themes that might enhance the depth and quality of the collected data from the next interviews. 

### 4.3. Data Analysis

A thorough analysis was conducted by reviewing the interview transcripts iteratively to validate the established themes and codes [83,84]. Specifically, all three authors collaboratively developed a codebook after an initial analysis of four interviews, each conducting a preliminary individual analysis before meeting to compare interpretations and ensure consistency. This iterative process refined the codebook, capturing the depth of the data while incorporating diverse perspectives and enhancing analytical rigour. The data analysis process was enhanced with the use of NVivo software (version 14), a robust qualitative data analysis tool, to streamline and systematise the coding process among researchers while also ensuring consistency in coding. Thematic analysis was employed, involving assigning symbolic meaning to the text and developing themes to capture recurring patterns [84].

The analysis identified two main themes (Figure 2). The first theme focused on staff perspectives on domestic violence, including their views on the escalation and severity of domestic violence, the reasons for the increase in VAW, and gaps in the legal framework. The second theme explored the challenges and experiences of professionals during the quarantine period, such as transitioning to virtual delivery, limited mobility and funding, supporting victims with limited shelter capacity, the impact on staff training and outreach work, and the well-being of domestic violence professionals.

## 5. Results

### 5.1. Professionals’ Perspectives on Domestic Violence during Quarantine

#### 5.1.1. Escalation of Violence

During the pandemic, professionals observed a significant rise in domestic violence cases, with many victims turning to shelters and hotlines for assistance. Participants noted that, “the situation with domestic violence worsened during the pandemic” (Participant 1), “more women were suffering from domestic violence” (Participant 9), and that they “received many calls” (Participant 8). Most participants reported a sustained increase in domestic violence, which continued throughout the first year of the lockdown and into the subsequent year of quarantine:

I do not know what it is connected with, but now we have more calls. (Participant 4)

We have observed an increase in the number of women seeking our assistance recently. Although there was a significant rise in help-seeking from domestic violence victims during the pandemic, it appears that the demand for support has not diminished. (Participant 10)

#### 5.1.2. Severity of Violence

Professionals were particularly concerned about the increased severity of violence against women who were living in abusive relationships prior to the pandemic: 

If a woman experienced abuse before the pandemic, the frequency and severity of her abuse increased during the pandemic. (Participant 7)

The severity of domestic violence consequences was exemplified by several participants, who noted extensive physical injuries of service recipients and the gravity of their condition. For example, one of the interviewed professionals shared that, “During quarantine, we accepted a woman who was severely beaten. I was shocked when she undressed. Her entire body was covered with bruises” (Participant 1).

#### 5.1.3. Reasons for Escalation in Violence

The continuous lockdown forced women to live in close proximity to their abusers, leading to increased isolation and enhanced surveillance, which left them vulnerable to domestic violence:

During the pandemic, couples who were previously separated due to work commitments found themselves confined to the same space. (Participant 6)

Professionals further attributed the increase in domestic violence during quarantine to worsened socioeconomic conditions in families. Lockdown measures led to issues such as unemployment and financial difficulties, including difficulties with loan repayments, which disrupted family harmony and increased the frequency of domestic violence. One participant shared the story of a service recipient to highlight the effects of quarantine and socioeconomic instability:

During the quarantine, her husband, who used to work and come home tired, found himself unemployed and witnessing the distress of his hungry children. Unable to handle the situation, his pride prevented him from seeking state assistance meant for children. This pressure and frustration ultimately manifested in mistreatment towards his partner. (Participant 4)

Professionals further associated the rise in domestic violence with alcohol abuse, which they attributed to the abusers’ loss of income: 

Husbands would often turn to drinking as a result of their low income or lack of employment. When they did have some money, they would often spend it on alcohol. Unfortunately, when they returned home drunk, they would resort to physically assaulting their wives, often in front of their children. (Participant 19)

#### 5.1.4. Gaps in Legal and Law Enforcement Framework

Amidst the escalation of violence, participants expressed concerns about the effectiveness of domestic violence laws in Kazakhstan, highlighting their limitations in supporting women’s decision to leave an abusive relationship. The dependency on the abuser was seen as a barrier to seeking help and accessing the necessary support:

Even the laws in Kazakhstan are not without their flaws. In contrast with other countries, such as Germany, where an abuser can be removed from the family home, in our country, it is often the abuser who owns the property, and the victim is the one compelled to leave. This leaves her with no safe place to go. As a result, even well-intentioned police officers may find it challenging to provide comprehensive assistance to the victim. (Participant 42)

The participants stressed the need to revise the law on combating domestic violence in Kazakhstan. They felt that the current law primarily focuses on assisting the victims but does not adequately address the actions of the abusers. A professional highlighted this concern regarding legal issues in the following quote:

Amending the law to combat domestic violence is imperative in the Republic of Kazakhstan. The current approach primarily focuses on assisting the victims, but it overlooks the crucial aspect of working with aggressors. Consequently, this perpetuates a cycle of repeated violence, as victims often find themselves returning to the same abusive environments. (Participant 14)

In addition to shortcomings in the legal framework, the majority of the participants expressed dissatisfaction with the work of police departments, citing victims’ fear of reporting domestic violence and the police departments’ inconsistent handling of such cases: 

Women are afraid to go to the police because the police say that they won’t file a criminal case for a domestic issue. This perpetuates a cycle of violence and leaves the victim trapped in an abusive situation. (Participant 11)

The professionals explained that the challenges faced by domestic violence victims were worsened by both weaknesses in legal protection and women’s limited awareness of their rights. Victims often endured domestic violence until it became severe enough for them to seek help from VAW organisations. It is important to note that during the pandemic, domestic violence professionals experienced increased dissatisfaction with law enforcement due to a surge in individuals seeking help. With the rising number of domestic violence incidents, there was a heightened need for improved monitoring of protective order enforcement, as professionals criticised the prevailing legal framework for its inadequacy in protecting victims:

There is no clear mechanism for assisting victims, monitoring the enforcement of protective orders, and measures to bring the abuser to justice. (Participant 40)

In summary, VAW organisations faced overwhelming demand from victims throughout the year-long quarantine. The extended lockdown increased vulnerability to violence due to heightened isolation and surveillance. Socioeconomic factors, such as unemployment, financial hardships, and alcohol abuse, contributed to the surge in domestic violence cases. Weak legal and law enforcement systems, coupled with limited awareness among victims about their rights, posed significant challenges for domestic violence victims during the pandemic.

### 5.2. Challenges and Adaptation Experiences

#### 5.2.1. Virtual Mode of Service Provision

Professionals from different organisations reported successful adaptation to quarantine measures through virtual service delivery. However, certain services were difficult or impossible to offer remotely. Consequently, despite increased demand for VAW services during the pandemic, anti-pandemic measures resulted in a decrease in the number of services and support provided to victims compared to pre-COVID-19 times. Many professionals faced challenges in providing services to women with special needs. Training and employment for service recipients became challenging to organise. For example, a participant noted that, “If the training was held online, they continued. Those, organised entirely in offline mode, were suspended for 3 or 4 months” (Participant 9). Some organisations could not offer shelters due to the complete transition to virtual work, while other organisations limited their capacity for new residents. However, some organisations continued to provide services, adapting to quarantine requirements. 

An array of other challenges faced due to the virtual mode of service provision were also reported by professionals:

Correctional work with the aggressors was suspended entirely and only conducted through phone and video calls, which proved to be ineffective. Legal and psychological services provided by crisis centre specialists were provided online, but they were found to be inefficient. (Participant 25)

Working online with documents related to hospitals, residency registration, and other services that transitioned to online access was challenging. There was a time when access to these services was completely unavailable, which added to the difficulty. (Participant 5)

As seen, the first quote highlights the limitations and inefficiencies of virtual interactions in correctional work, while the second emphasises practical difficulties and service disruptions during the transition to digital platforms. Both perspectives reveal the need for more robust solutions and support systems to manage such transitions effectively. 

While each crisis centre faced unique challenges during the lockdown, all participants recognised that the government’s preventative measures had an impact on all services. The main challenge was the professionals’ inability to provide comprehensive assistance to all domestic violence victims and support survivors with multiple needs. The pandemic profoundly affected the ability of organisations to effectively support domestic violence victims despite the transition to remote service delivery, prompting the need for more robust solutions and support systems to manage such transitions effectively.

#### 5.2.2. Restricted Movement

During the lockdown, the restriction on free movement significantly impacted the ability of professionals to provide comprehensive services and reach survivors. The lack of preparedness of VAW organisations and the government in responding to the challenges brought about by the public health emergency was evident in the absence of well-defined protocols during emergencies. Specifically, participants emphasised the difficulties they encountered due to checkpoints and roadblocks, which impeded their ability to reach victims who lived in areas that were geographically distant from the crisis centres:

It was extremely challenging when they initially implemented the city-wide blockade, which prohibited commuting. Entering the city became restricted, making it even more difficult to handle situations when a woman from a rural area reached out for help. If the government had allowed us to move freely, our jobs would have been easier. Despite serving the state on the borders, we were not granted permission to return home. Consequently, we were unable to reach families where instances of violence had been reported. (Participant 28)

This challenging situation required collaboration across different departments, including law enforcement, in order to address the difficulties faced by survivors and meet their needs effectively:

During that time, we closely collaborated with the police, specifically with special departments that work with women. Their assistance was invaluable, and we also consulted with them over the phone. However, it was a challenging situation. (Participant 30)

The lack of preparedness resulted in difficulties in reaching victims who lived in geographically distant areas. Collaboration across different departments, including law enforcement, proved vital to effectively address the needs of survivors.

#### 5.2.3. Restrained Budget and Understaffing

Another substantial challenge, especially in NGO organisations, was a lack of funding during the pandemic, resulting in reduced service delivery and staff numbers, thereby worsening the difficulties faced by survivors. The financial constraints faced by private and NGO organisations, which constitute the majority of VAW organisations in Kazakhstan, were evident in the challenges of adequately compensating staff, often resulting in some professionals working without remuneration. Several participants even shared instances where staff members volunteered their time and services:

Between August and October 2020, our organisation secured funding from an international organisation to provide consulting assistance to victims. This funding specifically enabled us to cover the salaries of a social worker, a lawyer, and a psychologist. However, even after the project concluded, we have continued to offer consulting assistance on a voluntary basis. (Participant 21)

Due to understaffing, an organisation had to rely heavily on volunteers for service delivery, which was partly due to financial constraints faced during the COVID-19 pandemic: 

I operate independently at the centre without any additional staff members. Due to the low salary offered, a psychologist recently resigned from her position. As a result, when I require the services of a psychologist, I rely on the kindness and generosity of volunteer psychologists who graciously offer their assistance free of charge. (Participant 5)

Professionals highlighted the importance of volunteers in staying connected with victims who had previously sought help, especially as the number of victims contacting the centre increased:

Volunteers played a crucial role in our efforts to maintain contact with victims who had reached out to us previously. We wanted to ensure their safety, understanding that they might not have the chance to call us when their husbands were in close proximity. (Participant 6)

Funding constraints amidst high demand for services challenged private and NGO-managed organisations, increasing their reliance on volunteers.

#### 5.2.4. Limited Shelter Capacity

The organisations grappling with the inability to cover essential expenses also experienced reduced shelter capacities. For example, according to Participant 13, their organisation had to cutoff the occupied area by 80 square meters, resulting in a significant decrease in the number of shelter places available. The closure of state VAW centres during the initial quarantine period posed a significant obstacle for domestic violence professionals in providing shelter for those in need, exacerbating the challenges faced by victims:

It was also difficult to provide shelter, as all the state centres did not work. (Participant 31)

Despite the suspension of some services during the quarantine, organisations made commendable efforts to assist survivors by finding shelters through various means. Participant 36 highlighted the challenges faced due to limited budgets and full shelters, which forced them to refer survivors to other organisations where vacancies were still available. Recognising the increased demand for shelters and the closure of local crisis centres during the quarantine, one organisation that previously catered to specific target groups, such as women in their third trimester of pregnancy or those with newborns, extended its services to include all women experiencing violence, regardless of their specific characteristics. Professionals prioritised maintaining communication with every woman seeking assistance, even if they were unable to receive direct support from the crisis centre.

#### 5.2.5. Staff Training and Outreach Work

During the emergency, domestic violence professionals needed to be prepared and trained to work online and follow anti-pandemic guidelines. However, only half of the participating professionals had access to such training. The implementation of preventative measures seemed to limit the ability of organisations to provide training for their staff: 

Conducting training for employees of sociopsychological support services for domestic violence victims, crisis centres, and representatives of police departments has become increasingly challenging due to the anti-epidemic restrictions. (Participant 45)

Additional challenges faced beyond staff training included the disruption of information dissemination work in educational sessions due to closures, as well as restricted outreach work in health institutions to prevent the spread of the virus. While outreach work became essential in raising public awareness about domestic violence and services available to protect victims during the pandemic, the organisations were unable to continue this service throughout the pandemic:

The outreach work faced challenges, as some organisations were closed, and medical organisations did not allow us to inform the population about the services due to virus prevention measures. (Participant 22)

During the pandemic, organisations encountered difficulties in training staff, disseminating information, and conducting outreach in health institutions.

#### 5.2.6. Health Concerns of Domestic Violence Professionals

As the professionals navigated the demands of their roles in restrictive conditions, their mental and physical health was significantly affected. Since not all services could be moved online, the provision of social care involved physical contact with domestic violence victims, which increased the risk of infection for staff members. For example, Participant 20 mentioned that all employees were infected while providing humanitarian help, and they were then in the process of receiving COVID-19 vaccines. Professionals shared their deep concern for the well-being of their families during the pandemic. 

Another professional mentioned the negative impact of sacrificing physical contact and face-to-face interaction with their own family. They had to switch to more expensive means of transport to minimise the infection risks for their family members and service recipients at the centre:

I was unable to have physical contact with my family. Their health and safety were my top priorities, so I chose not to risk their well-being by seeing them in person. Instead, I relied on phone calls to stay in touch. To minimise the risk of infection, I started taking taxis to work instead of buses. Buses were frequently crowded and posed a higher risk of infection. I understood that my own health was connected to the health of the children and mothers at the centre where I worked. (Participant 5)

Other stressors affecting the well-being of professionals included personnel reduction, reduced working hours for the remaining staff, and challenges with COVID-19 testing and vaccination for staff.

On the other hand, domestic violence professionals who transitioned to the online mode of service delivery felt lucky not to have infected staff members and service recipients. These participants believed that shifting to online services and following sanitary requirements helped them prevent the spread of the virus among staff and service recipients, while at the same time continuing their service:

Specialists transitioned to online mode, which proved to be beneficial in preventing the spread of the virus. (Participant 16)

All safety precautions were followed, and therefore, our employees were not infected with COVID. (Participant 27)

By following anti-pandemic measures, they were able to continue providing services, although in limited capacity, while also minimising the risks to their own health. 

## 6. Discussion

While violence can have severe impacts on women’s health and well-being, leading to an increased need for quality VAW services [85], the overarching narrative emerging from crisis studies is that violence or abuse, present prior to a national disaster, is likely to worsen during and after the crisis [15], challenging achievement of SDG 5.2 to eliminate VAW globally. 

VAW professionals in Kazakhstan also observed that victims who were experiencing domestic violence prior to the pandemic started reporting incidents to the police and seeking refuge in shelters as the violence and battering became severe. Several scholars have raised concerns about the pandemic exacerbating the severity of violence [19,25,35,36]. Professionals raised concerns about the dependency of women on their abusers, limitations in protective orders, and dissatisfaction with the response of law enforcement. These gaps, combined with women’s limited knowledge of their rights, contributed to the exacerbation of domestic violence in the country. In neighbouring Kyrgyzstan, the fear of reprisal from their husbands and worries about the well-being of their children compelled women to stay in abusive marriages [44]. 

Geographical challenges, such as restricted movement, checkpoints, and roadblocks, posed difficulties in reaching domestic violence victims, while outreach work in health centres was disrupted due to anti-COVID-19 measures. Kazakhstan’s vast rural areas became especially neglected, as VAW organisations are located in urban regions, leaving rural women with no immediate support and access to services. The disparity between rural and urban areas in terms of service availability and accessibility created additional layers of vulnerability for women in rural regions. 

Even in urban areas, VAW professionals in Kazakhstan faced challenges in providing support to victims and experienced insufficient shelter capacity for domestic violence victims. To address these challenges, VAW organisations in Kazakhstan had to provide online services under quarantine requirements, ensuring victims could still access the necessary assistance, as seen in VAW organisations worldwide [27,29,31,32,33,35]. However, many Kazakhstani VAW organisations also faced the additional hurdle of insufficient funding, resulting in limited service delivery and staff reductions. The strain on resources further compounded the difficulties faced by VAW organisations in meeting the increasing demands for support during the pandemic, as the number of domestic violence cases sharply increased. In addition to these constraints, the professionals themselves experienced social distancing requirements, heightened infection risks, and stress associated with adapting to new working conditions. 

The decriminalisation of battery in 2017 had profound implications for domestic violence cases. This legislative change removed the possibility of prosecuting many cases, which became particularly evident during the pandemic. Considering that the main perpetrators of domestic abuse are husbands or male intimate partners, these legal reforms were indicative of a patriarchal perspective that failed to prioritise the safety of women and reduced the state’s responsibility to hold perpetrators accountable. The lack of legal and police intervention reflects a broader societal reluctance to challenge patriarchal attitudes that overlook or normalise domestic violence. This issue is further compounded by the cultural concept of “uyat” (shame), which still discourages victims from seeking help. The pandemic only underscored these legal gaps, which created additional barriers for women seeking justice and limited the effectiveness of VAW organisations. 

In response to the growing recognition of these challenges, a new law on the prevention of domestic violence and the improvement of women’s safety was adopted in 2024 [80]. Indeed, this is an important step forward to strengthen legal protections for survivors, contribute to advancing gender equality (SDG 5) and promote good health and well-being (SDG 3). However, despite the positive developments [68], including the reclassification of domestic violence as a criminal offence [70,80], there remains a need for a more comprehensive approach to address SDG 5.2 effectively. Findings from times of emergency reveal that the role of crisis centres has not yet been fully integrated into the national discourse on SDG 5.2. To achieve the objectives of SDG 5.2, the government must enhance its support for crisis centres, ensuring they are adequately resourced during both emergency and non-emergency periods. This support is essential to protect domestic violence victims, who are often marginalised by existing laws and entrenched sociocultural norms.

Despite the COVID-19 pandemic no longer being in the global spotlight, VAW continues to be a pervasive issue. To effectively advance SDG 5.2, it is essential to implement legal changes and strengthen law enforcement, as well as tackle sociocultural challenges. Moreover, enhancing preparedness for future emergencies is critical for fostering sustainable change and ensuring that efforts to combat VAW are resilient and impactful.

## 7. Conclusions

During the COVID-19 pandemic, VAW organisations in Kazakhstan faced challenges in effectively operating due to disruptions in service provision and funding issues. The government’s neglect to place adequate emphasis on the VAW sector in its COVID-19 responses worsened the existing situation, leading to inequities and inefficiencies in VAW service provision. These actions hindered the promotion of gender equality and human rights but also affected the fulfilment of broader international commitments set forth in SDG 5.2, similar to their counterparts worldwide [26,28,29,30]. These lessons are essential for future crises, especially in the context of VAW. 

This study provided a nuanced perspective on domestic violence service provision during the pandemic in a Central Asian country, contributing to the global knowledge on domestic violence in crises. The findings highlighted the need to understand the impacts of emergencies, allocate sufficient resources, and implement effective policies addressing inequality and violence against women. Gender-sensitive initiatives, as seen in the COVID-19 Global Gender Response Tracker [1], should be integrated into recovery efforts and policymaking.

In addition, the study provided fresh perspectives on the mistreatment of women seeking assistance from VAW organisations during emergencies. It highlighted the need for additional support and resources beyond the crisis period in VAW organisations. Furthermore, the examination of adaptations made by VAW organisations during the pandemic also provided valuable insights for government, international agencies, and social care organisations combating VAW. To strengthen universal health coverage and access to quality health services in line with SDG 3.8, it is essential to foster adaptive responses to SDG 5.2. This will also help improve the effectiveness of interventions aimed at combating VAW during times of crisis and beyond.

### 7.1. Limitations

Despite its contributions, this study had some limitations. It focused on the experiences of VAW organisations in Kazakhstan, which may not fully capture the diversity of challenges faced in other Central Asian countries with different sociopolitical and sociocultural contexts. Furthermore, the viewpoints of survivors, law enforcement officials, and policymakers were not included, limiting the scope of implications offered.

Future research might investigate how other types of emergencies, such as natural disasters, military conflicts, economic downturns, and political crises, impact VAW organisations and the role of international aid and partnerships in supporting VAW services during crises. Additionally, it could explore how VAW organisations align their work with SDGs 5.2 and 3.8 and the extent to which these goals are integrated into their practices.

### 7.2. Implications

#### 7.2.1. Increase Funding for VAW Organisations 

The COVID-19 pandemic has highlighted the critical need for financial support for VAW centres to ensure they have the necessary resources to continue providing services during times of crisis.

#### 7.2.2. Improve Digital Infrastructure

Due to the pandemic, many VAW organisations had to switch to remote and online services. Reliable internet connectivity, secure data management systems, and virtual communication tools are crucial to better assist survivors during and after emergencies.

#### 7.2.3. VAW Professionals’ Training

It is crucial to raise awareness among VAW organisations and the community, more generally, about women’s vulnerability to domestic violence during emergencies [86] and train staff to better assist victims during crisis times [87]. Although health professionals play a crucial role in identifying and managing cases of violence, their training is often inadequate [88]. Agencies must have pre-emergency preparation and planning in order to effectively respond to protect women from domestic violence in disasters and emergencies [23,39]. 

#### 7.2.4. Increase Public Awareness

Educational and outreach programmes should be implemented in schools, colleges, and universities to raise awareness about the negative impacts of domestic violence and promote healthy relationships. Engaging men and boys in these programmes is also crucial in creating a shift towards non-violent relationships.

#### 7.2.5. Strengthen Legal Protections

It is crucial for Kazakhstan and other countries to enhance the legal framework aimed at safeguarding women and promoting awareness about existing laws among the general public. Measures to achieve this can involve providing legal aid and support services to survivors, increasing penalties for domestic violence offences, and implementing measures to enforce protection orders. 

#### 7.2.6. Context-Specific Interventions

Tailoring VAW services to the cultural, linguistic, ethnic, and religious contexts of Kazakhstan is essential for effectively addressing the diverse challenges VAW organisations encounter. Understanding local sociocultural norms and providing resources that can accommodate women’s (Kazakh or Russian) linguistic backgrounds can enhance outreach strategies and the accessibility of VAW services for women. Collaborating with religious and community leaders can be an effective strategy to combat VAW, as it enables the use of religious and community values to promote gender equality and challenge harmful norms that can negatively impact women.

## Figures and Tables

**Figure 1 ijerph-21-01326-f001:**
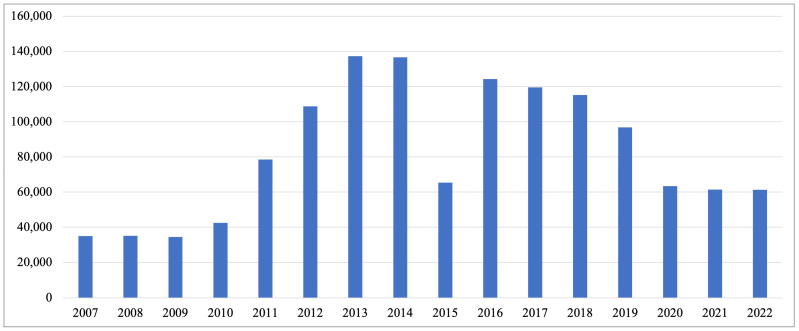
Reported incidences of VAW in Kazakhstan (2007–2022). Source: “Incidence of violence (recorded offences) against women during past period” (Bureau of National Statistics. (n.d.). Agency for Strategic Planning and Reforms of the Republic of Kazakhstan). Available online at: https://gender.stat.gov.kz/page/frontend/detail?id=85&slug=-70&cat_id=4&lang=en (accessed on 15 September 2024).

**Figure 2 ijerph-21-01326-f002:**
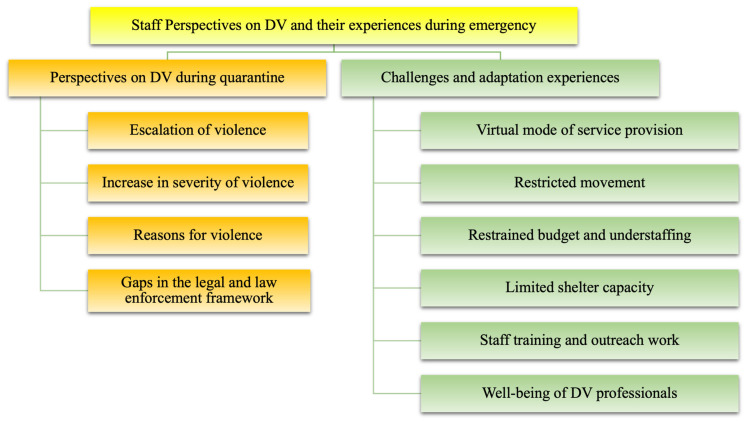
Professionals’ perspectives on domestic violence and their experiences during the COVID-19 emergency.

## Data Availability

Data cannot be shared publicly because the authors’ Institutional Research Ethics Committee has prohibited the public publication of the data.

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
