# Peer review of "Lessons Learned from the Experiences of Domestic Violence Service Providers in Times of Crisis: Insights from a Central Asian Country"

_ijerph, 2024, doi:10.3390/ijerph21101326_

Round 1
Reviewer 1 Report
Comments and Suggestions for Authors
Summary:
I believe this is an extremely pertinent article, which furthers existing knowledge in an area of study that demands further analysis. Regarding the research that was undertaken, as strong points, we can highlight the article’s focus on COVID-related circumstances that made VAW interventions even more difficult to carry out. Another strong point is that the authors also try to consider the hardships faced by the providers of these services and the different ways the pandemic took a toll on their resilience and ability to cope. The fact that this study focuses on a geographic region where VAW presents high prevalence makes it even more relevant to the issue at stake.
General comments
The article contributes to confirming our empirical perception of the negative effects of the pandemic on the exacerbation of VAW, thus joining some studies already published on the issue in other countries (like Viero et al 2021 and Barbara et al 2020, among others, some of which were mentioned by the authors). The literature review is adequately covered. The bibliographical references are adequate and updated. The manuscript is very clear and easy to read, and the sections are well-structured. Good overall scientific quality, with adequate methodology, which is properly described. The chart provided is clear and helps visualise the argumentation. The conclusions are supported by the analysis undertaken, although I believe that they could be further developed (please see below)
DETAILED COMMENTS
line 113-114: the first research question described here does not exactly match the first research question in 2. Materials and methods (line 127). Please revise.
line 144: Please clarify what is meant by ‘Mom’s Houses’.
line 422: Should it be ‘now’? Or rather ‘then’?
lines 412-414: I believe that this sentence properly belongs in the discussion section:
‘This emphasises the significance of using technology for virtual educational sessions and remote outreach while also modifying strategies to prioritise the safety of staff and the community.’
lines 448-449: I believe that this sentence properly belongs in the discussion section:
‘However, our analysis reveals that as the well-being of domestic violence professionals increased, the number and quality of services provided to survivors decreased.’
CONCLUSIONS
This is the only part of the article I feel could be substantially improved. It does highlight its contributions to the existing state-of-the-art, but it should also dwell on its limitations and possible ways to overcome them. This conclusion could also be improved by making specific suggestions on how future research could stem from this one's findings.
Author Response
Comment - Summary:
I believe this is an extremely pertinent article, which furthers existing knowledge in an area of study that demands further analysis. Regarding the research that was undertaken, as strong points, we can highlight the article’s focus on COVID-related circumstances that made VAW interventions even more difficult to carry out. Another strong point is that the authors also try to consider the hardships faced by the providers of these services and the different ways the pandemic took a toll on their resilience and ability to cope. The fact that this study focuses on a geographic region where VAW presents high prevalence makes it even more relevant to the issue at stake.
General comments
The article contributes to confirming our empirical perception of the negative effects of the pandemic on the exacerbation of VAW, thus joining some studies already published on the issue in other countries (like Viero et al 2021 and Barbara et al 2020, among others, some of which were mentioned by the authors). The literature review is adequately covered. The bibliographical references are adequate and updated. The manuscript is very clear and easy to read, and the sections are well-structured. Good overall scientific quality, with adequate methodology, which is properly described. The chart provided is clear and helps visualise the argumentation. The conclusions are supported by the analysis undertaken, although I believe that they could be further developed (please see below)
Response: Thank you for your thoughtful engagement with our paper and the constructive feedback you provided. We sincerely hope that our revisions have effectively addressed your detailed suggestions. For your convenience, we have highlighted the incorporated revisions in yellow in response to your comments, as well as those from the other two reviewers. Additionally, we have added new sections and several new references, which are also highlighted in yellow.
DETAILED COMMENTS
1: line 113-114: the first research question described here does not exactly match the first research question in 2. Materials and methods (line 127). Please revise.
Response: The first research question in the Introduction has been adjusted to align with the one that appears in Materials and Methods.
2. line 144: Please clarify what is meant by ‘Mom’s Houses’.
Response: The term “Mom’s Houses” was corrected to “Mother’s Houses”, and a clarification is provided when the term first appears in the text. These Mother’s Houses, located across the country, aim to support pregnant women and women with newborn infants, many of whom are also survivors of domestic violence.
3. line 422: Should it be ‘now’? Or rather ‘then’?
Response: Yes, “now” has been changed to “then”.
4: lines 412-414: I believe that this sentence properly belongs in the discussion section:
‘This emphasises the significance of using technology for virtual educational sessions and remote outreach while also modifying strategies to prioritise the safety of staff and the community.’
Response: This argument has been moved to the Discussion section, as suggested.
5: lines 448-449: I believe that this sentence properly belongs in the discussion section:
‘However, our analysis reveals that as the well-being of domestic violence professionals increased, the number and quality of services provided to survivors decreased.’
Response: This argument has been moved to the Discussion section, as suggested.
6. CONCLUSIONS
This is the only part of the article I feel could be substantially improved. It does highlight its contributions to the existing state-of-the-art, but it should also dwell on its limitations and possible ways to overcome them. This conclusion could also be improved by making specific suggestions on how future research could stem from this one’s findings.
Response: Thank you for your comment. The Conclusion section has been elaborated according to the comments from all reviewers. Implications for future research are also provided.
Reviewer 2 Report
Comments and Suggestions for Authors
Thank you for the valuable work and for sharing the experiences of the service providers in your region with the rest of the world. Your work is a valuable lesson for everyone who is passionate and engaged in the field of intimate partner violence and women empowerment. The issue of VAW is an ongoing challenge for women’s health and well-being and is highly impacted by the stability of the environment in which women's lives (like war or any social disruptions), along with individual and community factors like SES, culture, policies, religion, and ethnicity. Although we passed the COVID-19 crisis; however, the risk of emerging new infectious diseases, natural disasters, or humanitarian conflicts is inevitable. The lesson shared in this study and the recommendations provided can be useful globally and help policymakers and service providers to better planning for future possible crises.
Some suggestion:
This study has a particular setting, focusing on a specific Central Asian country, and many readers may not have enough information about this setting (including the country’s demographics, ethnicity, religion, culture, existing resources, and limitations). As a result, providing more detailed background information is necessary to provide a vivid picture of the context of this study. Moreover, knowing the background of existing organizations in Kazakhstan, policies related to VAW, and also the COVID-19 restrictive measures in this country would support the readers to better relate to the findings of this study and effectively adapt the study recommendations.
During this study, you moved between the terms VAW and IPV several times [lines 29-30 /113-114/ 127/128]. The term VAW defines a broad range of violence against women, and IPV is only one type of VAW. From the last part of your introduction (line 93 to the end), it seems the problem in your targeted country is mostly IPV and not other forms of VAW like human trafficking. I would suggest providing definitions for GBV, VAW, and IPV and clarifying which one you will use. It could be helpful to let the readers know what type of VAW is dominant in this country and why you only focused on IVP.
In your method section, provide more information about your sampling method. Did you use purposeful sampling or snowball sampling [Line 149-150]? You also can provide more details on the data collection and analysis setting (examples: who conducted the interviews? Were the interviews structured, semi-structured, or open? Who coded the data, and who conducted the thematic analysis?) This information is necessary to understand the setting in which data is collected and analyzed, the people involved in data analysis, and possible biases in each stage.
In the results (lines 290-299 which you talked about challenges), adding more direct quotes from your participants would be helpful.
In the discussion, lines 453-455, you specifically talked about aggravated violent behavior in a male with a history of IPV; please add a citation for that.
Adding a limitation paragraph to this study would be helpful.
For the conclusion, it will help if you remind the readers of the study recommendations and lessons learned in a short summary (or bullet points). Especially as you mentioned “context-specific interventions in Central Asia,” it would be great if you reiterated your context-specific suggestions.
Thank you for your valuable work.
Author Response
Comments and Suggestions for Authors
Thank you for the valuable work and for sharing the experiences of the service providers in your region with the rest of the world. Your work is a valuable lesson for everyone who is passionate and engaged in the field of intimate partner violence and women empowerment. The issue of VAW is an ongoing challenge for women’s health and well-being and is highly impacted by the stability of the environment in which women’s lives (like war or any social disruptions), along with individual and community factors like SES, culture, policies, religion, and ethnicity. Although we passed the COVID-19 crisis; however, the risk of emerging new infectious diseases, natural disasters, or humanitarian conflicts is inevitable. The lesson shared in this study and the recommendations provided can be useful globally and help policymakers and service providers to better planning for future possible crises.
Response: Thank you for your thoughtful engagement with our paper and the constructive feedback you provided. We sincerely hope that our revisions have effectively addressed your detailed suggestions. For your convenience, we have highlighted the incorporated revisions in yellow in response to your comments, as well as those from the other two reviewers. Additionally, we have added new sections and several new references, which are also highlighted in yellow.
Some suggestion:
1. This study has a particular setting, focusing on a specific Central Asian country, and many readers may not have enough information about this setting (including the country’s demographics, ethnicity, religion, culture, existing resources, and limitations). As a result, providing more detailed background information is necessary to provide a vivid picture of the context of this study. Moreover, knowing the background of existing organisations in Kazakhstan, policies related to VAW, and also the COVID-19 restrictive measures in this country would support the readers to better relate to the findings of this study and effectively adapt the study recommendations.
Response: Thank you for your comment. In response, we have added a section on the Research Context to the article (pp. 3-5). This section covers two key areas: (1) the broader Central Asian context, focusing on the concept of “shame,” which significantly influences social morals, ethics, and gender dynamics, and (2) the development of Kazakhstan’s legislative framework for domestic violence, emphasising the socio-political factors at play. Additionally, we included a brief overview of organisations addressing violence against women (VAW) in Kazakhstan and discussed the impact of the pandemic lockdown on domestic violence to further contextualise the issue.
2. During this study, you moved between the terms VAW and IPV several times [lines 29-30 /113-114/ 127/128]. The term VAW defines a broad range of violence against women, and IPV is only one type of VAW. From the last part of your introduction (line 93 to the end), it seems the problem in your targeted country is mostly IPV and not other forms of VAW like human trafficking. I would suggest providing definitions for GBV, VAW, and IPV and clarifying which one you will use. It could be helpful to let the readers know what type of VAW is dominant in this country and why you only focused on IVP.
Response: This comment has been addressed in several sections of the paper, while we still retain both the terms “VAW” and “domestic violence” for specific reasons. The term “VAW organisations” indicates a broader focus on supporting women experiencing various forms of violence. However, our study specifically examines domestic violence in the country. It explores the perspectives of professionals working in VAW organisations regarding the increase in domestic violence during the pandemic and their experiences in supporting survivors.
We have also added a new section titled “Conceptual and Terminological Overview,” where we discuss the terms “VAW,” “domestic violence,” and “intimate partner violence.” Given the context of Kazakhstan, we prioritise the term “domestic violence” over “intimate partner violence” because it is more relevant, especially in cases where violence is influenced by interactions between a woman’s husband and her mother-in-law.
Additionally, the research context section highlights that domestic violence is the predominant form of VAW in Kazakhstan.
3. In your method section, provide more information about your sampling method. Did you use purposeful sampling or snowball sampling [Line 149-150]? You also can provide more details on the data collection and analysis setting (examples: who conducted the interviews? Were the interviews structured, semi-structured, or open? Who coded the data, and who conducted the thematic analysis?) This information is necessary to understand the setting in which data is collected and analysed, the people involved in data analysis, and possible biases in each stage.
Response: We have expanded the Materials and Methods section to address this comment. We added a detailed discussion of the sampling method. Additional information about each participant can be found in Appendix A. We also included more details about the instrument and data collection process. Furthermore, the sub-section on data analysis has been revised to clarify the analytical processes used by the researchers.
4. In the results (lines 290-299 which you talked about challenges), adding more direct quotes from your participants would be helpful.
Response: In response to this comment, we have expanded this section to include additional direct quotes from participants, which further illustrate the challenges they faced (see lines 413-443). We have also added further interpretation of these new quotes within this section.
5. In the discussion, lines 453-455, you specifically talked about aggravated violent behavior in a male with a history of IPV; please add a citation for that.
Response: The language of this sentence has been softened and supported with a citation.
6. Adding a limitation paragraph to this study would be helpful.
Response: A paragraph discussing the study’s limitations has been added. This section specifically highlights the narrow focus on VAW professionals in Central Asia. Incorporating the perspectives of survivors and law enforcement officers could further strengthen the study by providing additional insights and enhancing the overall depth of the research.
7. For the conclusion, it will help if you remind the readers of the study recommendations and lessons learned in a short summary (or bullet points). Especially as you mentioned “context-specific interventions in Central Asia,” it would be great if you reiterated your context-specific suggestions.
Response: The Conclusion section has been expanded based on the feedback from all reviewers. A brief summary and recommendations have been included to enhance this section of the manuscript (pp. 16-17, lines 665-696). Specifically, context-specific implications derived from the main arguments of the manuscript are detailed in lines 689-696.
7. Thank you for your valuable work.
Response: Thank you for your valuable feedback, which has been instrumental in helping us refine the manuscript.
Reviewer 3 Report
Comments and Suggestions for Authors
The article explores how service providers assess their capacity to help in DV cases during the COVID-19 pandemic in Kazakhstan. It is based on the interviews (45) and uses quality methods to extract information and make conclusions. The article certainly has some merits: it is a new material, coming from a crucial Central Asian country. However, it needs improvements in background, introduction, discussion and conclusions.
1) there needs to be a more detailed and analytical background on Kazakhstan (p. 3) - what is the state of domestic violence protection system in the country, how do the shelters and crisis centres work (p. 4 -description in organizations), are there any challenges for the NGOs considering the situation in Kazakhstan etc. - this shall help to contextualize the findings much better;
2) stats needs to be updated, the authors can use the latest report of the Ombudsperson on the state of DV in Kazakhstan - https://www.gov.kz/uploads/2024/3/1/5a7f4aa16b071a92334a67807af65ea9_original.9433178.pdf.
3) in description of methods, the table with the main characteristics of interviewees (age, sex, region, status of the organization, position, experience) should be added to better explain the findings and connect them with conclusions;
4) the authors mention SDGs 5.2 and 3.8, but never elaborate on how they are connected with the material (for example, do service providers think about SDGs, tailor their work, not reflect at all etc.) or with the situation in Kazakhstan (how much Kazakhstan is involved in performing on those SDGs);
5) the discussion reflects all the major problems service providers have all over the world: it would be good to see what is specific for Kazakhstan or, alternatively, how Kazakhstani service providers reflect on the issue of DV specifically in their region (in relation to culture, regional differences, ethnic differences - Kazakh vs Russian, for example, etc.);
6) Conclusions shall reflect on what it is that Kazakhstan case study can offer to the international scholarship except for the obvious.
Comments on the Quality of English LanguageThe authors need to do an additional copy-editing for typos and grammar.
Author Response
The article explores how service providers assess their capacity to help in DV cases during the COVID-19 pandemic in Kazakhstan. It is based on the interviews (45) and uses quality methods to extract information and make conclusions. The article certainly has some merits: it is a new material, coming from a crucial Central Asian country. However, it needs improvements in background, introduction, discussion and conclusions.
Response: Thank you for your thoughtful engagement with our paper and the constructive feedback you provided. We sincerely hope that our revisions have effectively addressed your detailed suggestions. For your convenience, we have highlighted the incorporated revisions in yellow in response to your comments, as well as those from the other two reviewers. Additionally, we have added new sections and several new references, which are also highlighted in yellow.
1) there needs to be a more detailed and analytical background on Kazakhstan (p. 3) - what is the state of domestic violence protection system in the country, how do the shelters and crisis centres work (p. 4 -description in organisations), are there any challenges for the NGOs considering the situation in Kazakhstan etc. - this shall help to contextualise the findings much better;
Response: In response to this comment, we have added a section on the Research Context to the article (pp. 3-5). This section covers two key areas: (1) the broader Central Asian context, with a focus on the concept of “shame,” which significantly influences social morals, ethics, and gender dynamics, and (2) the development of Kazakhstan’s legislative framework for domestic violence, highlighting the socio-political factors that impact this issue. Additionally, we provide a brief overview of organisations addressing violence against women (VAW) in Kazakhstan, as well as the effects of the lockdown on domestic violence, to further contextualise the discussion. This section clarifies the socio-political environment in which VAW organisations operate in Kazakhstan.
2) stats needs to be updated, the authors can use the latest report of the Ombudsperson on the state of DV in Kazakhstan - https://www.gov.kz/uploads/2024/3/1/5a7f4aa16b071a92334a67807af65ea9_original.9433178.pdf.
Response: The statistics were updated using the suggested source as well as other available materials from the Bureau of National Statistics (see Figure 1).
3) in description of methods, the table with the main characteristics of interviewees (age, sex, region, status of the organisation, position, experience) should be added to better explain the findings and connect them with conclusions;
Response: We have expanded the Materials and Methods section to address this comment. Details about each participant can be found in Appendix A. The table includes all the information suggested in your comment, except for age. While we cannot disclose the exact ages of the participants, we have included their years of experience to help fill this gap.
We recognise that linking the characteristics and specific details of each participant to the findings could provide a more nuanced understanding. However, our study does not focus on exploring regional, ethnic, or demographic differences related to position, age, or gender among participants, organisations, or domestic violence survivors. Additionally, it is important to note that the majority of participants were female (with the exception of three males), which limited our ability to observe potential gender differences in perceptions and experiences. This is an area that warrants further research.
4) the authors mention SDGs 5.2 and 3.8, but never elaborate on how they are connected with the material (for example, do service providers think about SDGs, tailor their work, not reflect at all etc.) or with the situation in Kazakhstan (how much Kazakhstan is involved in performing on those SDGs);
Response: This comment is addressed in the Discussion and Conclusion sections of the manuscript. In our discussion (lines 624-629), we note that the recent adoption of a new law on the prevention of domestic violence and the enhancement of women’s safety in Kazakhstan represents a significant advancement in legal protections for survivors. This initiative contributes to promoting gender equality (SDG 5) and improving health and well-being (SDG 3). However, the government’s insufficient focus on the violence against women (VAW) sector during its pandemic response has hindered progress in gender equality and human rights, ultimately affecting Kazakhstan’s ability to meet its commitments to SDG 5.2.
While our study did not specifically examine how service providers in Kazakhstan engage with the SDGs in their daily operations, it is clear that ongoing socio-cultural issues and the need for better preparedness for future emergencies are essential for fostering sustainable change. We acknowledge that further research could investigate how VAW organisations align their work with the SDGs and the extent to which these goals are integrated into their practices.
5) the discussion reflects all the major problems service providers have all over the world: it would be good to see what is specific for Kazakhstan or, alternatively, how Kazakhstani service providers reflect on the issue of DV specifically in their region (in relation to culture, regional differences, ethnic differences - Kazakh vs Russian, for example, etc.);
Response: While our primary aim was not to delve into cultural, regional, and ethnic differences in detail, we recognise the importance of these factors in understanding domestic violence (DV) service provision. Further research could contribute to this area by exploring how regional and ethnic factors influence the experiences and reflections of service providers on DV.
As this was not our aim, our data did not allow for an in-depth exploration of these differences; however, we have provided an explicit description of the socio-cultural context of Kazakhstan and Central Asia within the manuscript (see section Research Context on pages 3-5 (lines 126-243). This contextualisation helped situate our findings and acknowledge the unique challenges faced by Kazakhstani service providers, which we further elaborate on in the Discussion and Conclusion sections of the study.
6) Conclusions shall reflect on what it is that Kazakhstan case study can offer to the international scholarship except for the obvious.
Response: In response to this comment, we have elaborated on the Conclusion section to provide more explicit implications relevant to both Kazakhstani and international contexts. In particular, we have included a concise summary and specific recommendations that highlight the unique insights derived from Kazakhstan, aiming to inform and enrich international scholarship (pp. 16-17, lines 665-696).
- Comments on the Quality of English Language: The authors need to do an additional copy-editing for typos and grammar.
Response: Thank you. We have proofread the manuscript. Please note we have used British English.
Round 2
Reviewer 3 Report
Comments and Suggestions for Authors
This is a much improved version. I would still like to see more engagement with SDGs, but I guess this can be a minor thing.
Comments on the Quality of English LanguageThe language is clear and without major awkwardness.
Author Response
Comment 1: This is a much improved version. I would still like to see more engagement with SDGs, but I guess this can be a minor thing.
Response: Thank you very much for your satisfaction with the revisions made and your minor comment. We have added text in the Discussion and Conclusion Sections, which are highlighted in yellow for your attention.
Comment 2: Comments on the Quality of English Language: The language is clear and without major awkwardness.
Response. Thank you for all your advice and suggestions. We hope you are satisfied with the revisions made.